# Cholangiocarcinoma Malignant Traits Are Promoted by Schwann Cells through TGFβ Signaling in a Model of Perineural Invasion

**DOI:** 10.3390/cells13050366

**Published:** 2024-02-20

**Authors:** Valerio de Franchis, Simonetta Petrungaro, Elisa Pizzichini, Serena Camerini, Marialuisa Casella, Francesca Somma, Enrico Mandolini, Guido Carpino, Diletta Overi, Vincenzo Cardinale, Antonio Facchiano, Antonio Filippini, Eugenio Gaudio, Cinzia Fabrizi, Claudia Giampietri

**Affiliations:** 1Department of Anatomy, Histology, Forensic Medicine and Orthopedics, Sapienza University of Rome, 00161 Rome, Italy; valerio.defranchis@uniroma1.it (V.d.F.); simonetta.petrungaro@uniroma1.it (S.P.); elisa.pizzichini@uniroma1.it (E.P.); francesca.somma@uniroma1.it (F.S.); mandolini.1617846@studenti.uniroma1.it (E.M.); guido.carpino@uniroma1.it (G.C.); diletta.overi@uniroma1.it (D.O.); eugenio.gaudio@uniroma1.it (E.G.); cinzia.fabrizi@uniroma1.it (C.F.); 2Core Facilities, Istituto Superiore di Sanità, 00161 Rome, Italy; serena.camerini@iss.it (S.C.); marialuisa.casella@iss.it (M.C.); 3Department of Medico-Surgical Sciences and Biotechnology, Sapienza University of Rome, 04100 Latina, Italy; vincenzo.cardinale@uniroma1.it; 4Laboratory of Molecular Oncology, Istituto Dermopatico dell’Immacolata, IDI-IRCCS, 00167 Rome, Italy; a.facchiano@idi.it

**Keywords:** bile duct cancer, Schwann cells, TGFβ, EMT

## Abstract

The term cholangiocarcinoma (CCA) defines a class of epithelial malignancies originating from bile ducts. Although it has been demonstrated that CCA patients with perineural invasion (PNI) have a worse prognosis, the biological features of this phenomenon are yet unclear. Our data show that in human intrahepatic CCA specimens with documented PNI, nerve-infiltrating CCA cells display positivity of the epithelial marker cytokeratin 7, lower with respect to the rest of the tumor mass. In an in vitro 3D model, CCA cells move towards a peripheral nerve explant allowing contact with Schwann cells (SCs) emerging from the nerve. Here, we show that SCs produce soluble factors that favor the migration, invasion, survival and proliferation of CCA cells in vitro. This effect is accompanied by a cadherin switch, suggestive of an epithelial–mesenchymal transition. The influence of SCs in promoting the ability of CCA cells to migrate and invade the extracellular matrix is hampered by a specific TGFβ receptor 1 (TGFBR1) antagonist. Differential proteomic data indicate that the exposure of CCA cells to SC secreted factors induces the upregulation of key oncogenes and the concomitant downregulation of some tumor suppressors. Taken together, these data concur in identifying SCs as possible promoters of a more aggressive CCA phenotype, ascribing a central role to TGFβ signaling in regulating this process.

## 1. Introduction

Cholangiocarcinoma (CCA) is a class of tumors derived from the epithelial lining of the biliary tree, with increasing incidence in Western countries [1]. Based on their anatomical location, they are subdivided into intrahepatic (iCCA) and extrahepatic cholangiocarcinoma (eCCA: perihilar and distal CCA), with different pathogenesis and clinical management [2]. One of the most striking features of CCA is neurotropism. Perineural invasion (PNI), with tumor-infiltrating fibers of the surrounding nerve plexi, is reported in about 75% of cases. In CCA, PNI is associated with a worse prognosis and shorter survival in CCA patients, representing an independent prognostic factor, allowing one to predict a poor outcome; however, we still know little about the underlying cellular and molecular mechanisms [3]. In the last ten years, a new research field, named cancer neuroscience, has emerged to investigate contribution of the nervous system to cancer progression [4]. Schwann cells (SCs) are glial cells well known for their role in ensheathing peripheral nerve fibers. During nerve repair and tissue regeneration, they shift to an activated phenotype, which associates with the upregulation of specific markers, such as glial fibrillary acidic protein (GFAP) and c-Jun [5,6]. Recently, many experimental data obtained on different types of tumors have converged to identify SCs as key elements in regulating tumor progression. The presence of GFAP+ SCs has been documented in several malignancies [7,8,9], including pancreatic ductal adenocarcinoma (PDAC), a tumor that shares many similarities with CCA [10]. Studies investigating SCs in PDAC highlighted their pivotal role in PNI and cancer progression [11,12,13]. Gundlach and colleagues recently reported the association between the presence of SCs and a worse prognosis in CCA patients [14]. We hypothesize that soluble factors typically released by SCs during nerve repair could promote the aggressive traits of CCA in PNI. TGFβ is known to be released by SCs [15,16,17,18], and this cytokine has a well-characterized role in promoting wound healing and inducing epithelial–mesenchymal transition (EMT) [19]. By undergoing EMT, cells shift their phenotype from epithelial to mesenchymal by losing cell–cell junctions (via E-cadherin downregulation), reorganizing their cytoskeleton to acquire motility and gaining the capability to digest and invade the extracellular matrix. EMT is a relevant process in both physiology (reparative and developmental processes) and pathology, being one of the key events leading to tumor metastasis [20]. In the present work, we found a lower positivity for cytokeratin 7 in CCA cells surrounding and invading the nerves in histological sections from CCA patients, with respect to the tumor mass. In vitro, we demonstrate the pro-tumoral effects of human primary SC (hSC)-secreted factors on CCA cell lines (HuCC-T1 and Oz). Furthermore, through proteomic analysis, we identified several proteins differentially regulated in CCA cells when cocultured with hSCs. Lastly, we show that a specific TGFβ receptor 1 (TGFBR1) antagonist can revert the increased migration and invasion of CCA cells driven by the exposure to hSC-secreted factors.

## 2. Materials and Methods

### 2.1. Patients and Samples

CCA specimens were collected from N = 33 patients undergoing liver resection with curative intent at the Policlinico Umberto I Hospital (Sapienza University of Rome, Rome, Italy). According to the anatomical location, tumors were classified as intrahepatic CCA (iCCA; N = 33).

### 2.2. Histomorphology and Immunohistochemistry on Human CCA Samples

Specimens were fixed in 10% buffered formalin for 2–4 h and embedded in low-fusion-temperature paraffin (55–57 °C). Then, 3–4 μm sections were obtained. Sections were stained with hematoxylin and eosin (H&E) according to standard protocols. For immunohistochemistry, endogenous peroxidase activity was blocked by a 20′ incubation in hydrogen peroxide (4%). Antigens were retrieved, as indicated by the vendor, by applying proteinase K (code S3020, Dako, Agilent Technologies, Glostrup, Denmark) for 10′ at room temperature. Sections were then incubated overnight at 4 °C with anti-cytokeratin 7 antibody (code M7018, Dako, Agilent Technologies). Samples were rinsed twice with PBS for 5′ and incubated for 30′ at room temperature with Dako REAL EnVision Detection System (code K5007, Dako, Agilent Technologies) using diaminobenzidine as substrate. Finally, sections were counterstained with hematoxylin. For all immunoreactions, negative controls (primary antibody replaced by pre-immune serum) were also included.

### 2.3. Histopathological Assessment of Human CCA Samples

All stained sections were examined in a coded fashion by Leica Microsystems DM 4500 B Light and Fluorescence Microscope (Weltzlar, Germany), equipped with a Jenoptik Prog Res C10 Plus Videocam (Jena, Germany). Moreover, slides were scanned by a digital scanner (Aperio CS2, Aperio, Leica Biosystems, Milan, Italy) and processed by ImageScope. Histological assessment of tumor slides was performed on H&E-stained slides.

The number of nerves within the tissue slide and the number of nerves affected by PNI by the tumor were also evaluated on H&E stains. The positivity for cytokeratin 7 (CK7) was calculated by an image analysis algorithm. Specific regions of interest (ROIs), namely, tumor mass vs. tumor cell infiltrating nerves, were selected in ImageScope and analyzed. Positivity to CK7 is expressed as the percentage of positive area within ROI.

### 2.4. Cell Cultures and Conditioned Media Preparation

HuCC-T1 and Oz cell lines, derived from intrahepatic CCA (iCCA), were purchased from Japanese Collection of Research Bioresources Cell Bank (JCRB) and cultured in Dulbecco’s modified Minimum Essential Medium (DMEM) (Sigma-Aldrich, Milan, Italy), supplemented with 10% Fetal Bovine Serum (FBS), 2 mM glutamine, 100 μg/mL streptomycin and 100 U/mL penicillin. Oz was previously established by Prof. Seishi Nagamori [21]. Primary human Schwann cells (hSCs) were purchased from Sciencell Research Laboratories (Carlsbad, CA, USA) and cultured in Schwann cell medium (SCM), purchased from the same company, supplemented with 5% FBS, 100 μg/mL streptomycin, 100 U/mL penicillin and Schwann Cell Growth Supplements (SCGS) (Sciencell). hSCs were plated on poly-D-lysine hydrobromide (25 μg/mL) (Sigma-Aldrich)-coated surfaces. To prepare conditioned media (CM), 5 × 10^5^ hSCs, Oz or HuCC-T1 cells/well were seeded in 6-well plates and allowed to attach overnight. The day after plating, cells were washed with phosphate-buffered saline (PBS) (Sigma-Aldrich); then, 2 ml DMEM supplemented with 5% FBS, 2 mM glutamine, 100 μg/mL streptomycin and 100 U/mL penicillin was added to each well. After 48 h, CM was collected, centrifuged at 1600× *g* for 10′ and then kept on ice until use.

### 2.5. Nerve Explants and 3D Perineural Invasion (PNI) Assay

All procedures involving animals were carried out in accordance with the guidelines of the European Community Council Directive (86/609/EEC of 24 November 1986) and the Italian National law DL 26/14. Following the protocol described by Demir and colleagues [7], postnatal day 4–5 CD1 mouse pups were sacrificed, and sciatic nerves were quickly harvested under a horizontal-flow fume hood and placed in ice-cold PBS. Glass bottom, 20 mm Petri dishes (Wuxi NEST biotechnology, Woodbridge, NJ, USA) were marked with a lab marker for reference points; then, a 25 μL ECM gel (E1270, Sigma-Aldrich) drop was plated in the middle of each dish, and one sciatic nerve explant was included in it. After gel polymerization, 2 × 10^5^ HuCC-T1 cells were suspended in 25 μL ECM gel and then plated 6 mm from the center of the dish, on the right of the nerve gel drop. After polymerization, a 5 μL ECM gel bridge was pipetted in between the two gel drops. Three ml of culture medium was added, and the system was incubated at 37 °C and 5% CO_2_ for 7 days. Every 24 h, pictures of the system were taken in fixed points. At the end of the experiment, immunofluorescence stain with rabbit anti-GFAP polyclonal antibody (code Z0334, Dako, Agilent Tecnologies) and DAPI (Thermo Fisher Scientific, Rockford, IL, USA) was performed, and micrographs were taken under a fluorescence microscope (Eclipse E600; Nikon Instruments S.p.A., Firenze, Italy).

### 2.6. Migration and Invasion Assays

Migration and invasion assays were performed using 12-well transwell chambers (Corning, New York, NY, USA). HuCC-T1 cells (2 × 10^5^) or Oz cells (1.5 × 10^5^) were seeded in the upper chamber of 8 μm pore size insert in the 12-well plates, with (invasion assay) or without (migration assay) growth factor-reduced Matrigel (code 354230, Corning, New York, USA) precoating (Matrigel was diluted 1:1 in serum free medium). An FBS gradient was created across the transwell membrane by placing serum-free medium in the upper chamber and hSC-conditioned medium (CM) in the lower compartment. Control medium consisted of medium conditioned by HuCC-T1 or Oz cells themselves (hereafter control CM). Both hSC CM and control CM contained 1% FBS. After 16 h (HuCC-T1 cell migration), 20 h (Oz cell migration) or 24 h (HuCC-T1 or Ozinvasion), cells on the upper surface of the transwell membrane were removed with a cotton tip; then, membranes were fixed with 4% PFA for 20′. A wash in PBS for 15′ was made, and then membranes were stained with 400 nM DAPI for 5′. After a 5′ wash in PBS, the membranes were mounted in 70% glycerol onto microscope slides, and micrographs were taken under Eclipse E600 (Nikon Instruments S.p.A) fluorescence microscope, in 9 fixed high-magnification fields. The number of cells on the bottom surface of the membrane was counted using ImageJ software 1.8 (National Institute of Health, Bethesda, MD, USA), and a comparison was made between the mean number of cells in the hSC CM samples and the control CM samples. To test TGFβ involvement in migration and invasion of CCA cells, TGFβ receptor 1 (TGFBR1) antagonist SB431542 (code 1614/1, Bio-Techne, Tocris, Minneapolis, MN, USA) was added to conditioned media at 10 μM final concentration and kept for the entire duration of each experiment.

### 2.7. Cell Viability Assay

HuCC-T1 cells were seeded in triplicate at the bottom of 96-well plates, at two different cell densities: 1 × 10^4^ cells/well (conventional cell density) and 2.5 × 10^3^ cells/well (suboptimal cell density). We seeded HuCC-T1 at suboptimal density to look for any potential pro-survival effect of hSC CM that would rescue them from this stressing condition. After 24 h, cells were washed in PBS, then incubated with hSC CM or control CM for 48 h. For the last three hours of treatment, 10 μL/well MTT (3-4,5-dimethylthiazol-2-yl)-2,5-diphenyl tetrazolium bromide (Sigma-Aldrich) (5 mg/mL) was added. The experiment ended by lysing cells with acidic isopropanol (HCl 0.04 N); then, optical density measurement was taken using the spectrophotometer (550 nm filter). The mean cell viability of the cells treated with hSC CM was compared to the sample treated with control CM.

### 2.8. Clonogenic Assay

HuCC-T1 cells were seeded in 6-well plates (2.5 × 10^2^–1.5 × 10^3^ cells/well) and allowed to attach for 6 h. Cells were treated with hSC CM or control CM for 5 days. After 72 h of treatment, 10 μM 4-(2-hydroxyethyl)-1-piperazineethanesulfonic acid (HEPES) was added as a buffer. At the end of the experiment, cells were fixed with 6% glutaraldehyde and stained with 0.5% crystal violet. The number of colonies formed in the control CM-treated wells, divided by the number of cells seeded, was expressed as “plating efficiency (PE)”. For hSC CM-treated samples, the number of colonies generated was divided by the PE times the number of cells seeded following the protocol by Franken and colleagues [22].

### 2.9. Proliferation Assay (BrdU Incorporation)

HuCC-T1 or Oz cells were seeded (1 × 10^4^ cells/well) in 96-well plates. After 24 h, cells were washed in PBS, and then hSC CM or control CM was added. Cell proliferation was assessed by BrdU incorporation (cell proliferation ELISA, BrdU kit, code 11647229001, Roche Life Sciences, Penzberg, Germany) according to manufacturer’s instructions. Briefly, during the last 18 h before the end of each timepoint (24 and 48 h), 10 μL/well BrdU labeling solution was added (final concentration 10 μM). Labeling medium was removed, and then cells were dried and fixed with “FixDenat” solution (200 μL/well) for 30′ at room temperature. Cells were incubated 90′ with 100 μL/well anti-BrdU-POD working solution. After 3 washes with washing solution, 100 μL/well substrate solution was added. Optical density measurement was taken using the spectrophotometer (405 nm; reference 655 nm). The mean BrdU incorporation of the cells treated with hSC CM was compared to the sample treated with control CM.

### 2.10. Western Blotting and Differential Proteome

#### 2.10.1. Protein Extraction

HuCC-T1 or Oz cells (1.85 × 10^5^) were suspended in DMEM supplemented with 5% FBS, 2 mM glutamine, 100 U/mL penicillin and 100 μg/mL streptomycin and seeded in 12-well plates. For coculture experiments, CCA cells were placed in contact indirectly with 1.85 × 10^5^ hSCs through a 3 μm pore size transwell insert. Single cultures (without transwell inserts) were used as controls. After transwell insert removal and ice-cold PBS wash, cells were lysed in Lysis buffer (code 9803S, Cell Signaling Technology, Danvers, MA, USA) in the presence of 2% SDS and 1 mM phenylmethylsulphonyl fluoride (PMSF).

#### 2.10.2. Western Blotting

Proteins were separated by SDS–PAGE and transferred on nitrocellulose membranes (Amersham Bioscience, Piscataway, NJ, USA) by electroblotting. Membranes were probed using the following antibodies: anti-β-actin-HRP (code A3854, Sigma-Aldrich 1:10,000); anti-tubulin (code T6074, Sigma-Aldrich 1:10,000); anti-E-cadherin (code 3195S, Cell Signaling Technology); anti N-cadherin (code ab18203, Abcam, Cambridge, UK); anti-cytokeratin 7 antibody (code M7018, Dako, Agilent Technologies 1:2000). Secondary antibodies were horseradish peroxidase-conjugated anti-mouse or anti-rabbit (Bio-Rad, Hercules, CA, USA). Membranes were washed with Tris-buffered saline with 0.1% Tween-20 (Sigma-Aldrich) and developed through chemiluminescence system (Amersham Bioscience) on the ChemiDoc Image Analyzer (Bio-Rad). Image lab software was used for densitometric quantifications.

#### 2.10.3. Differential Proteome

Samples and controls were analyzed in quadruplicate. Proteins were loaded on a pre-cast acrylamide 4–12% gradient gel (NuPAGE, Thermo Fisher Scientific) and run for about 10′ at 200 V. The gel was stained with Coomassie (Imperial protein Stain; Thermo Fisher Scientific), and the blue area was excised, destained, treated for cysteine reduction and alkylation and digested with 12 ng/µL trypsin (Promega Corporation, Fitchburg, WI, USA) at 37 °C overnight. The resulting peptide solution (5 µL) was injected in an Ultimate 3000 UHPLC (Dionex, Thermo Fisher Scientific) coupled with an Orbitrap Fusion Tribrid mass spectrometer (Thermo Fisher Scientific). Peptides were desalted on a trap column (Acclaim PepMap 100 C18, Thermo Fisher Scientific) and then separated on a 45 cm long silica capillary (Silica Tips FS 360-75-8, New Objective, Littleton, MA, USA), then packed in house with a C18, 1.9 μm, 100 Å resin (MichromBioResources, Auburn, CA, USA). The analytical separation ran for 180′ using a gradient of buffer A (5% acetonitrile and 0.1% formic acid) and buffer B (95% acetonitrile and 0.1% formic acid). The chromatographic starting point was 5% of buffer B, which rose to 6% in 5′, to 32% in 130′, to 55% in 25′ and up to 80% in 4′. Full-scan MS data were acquired in the 350–1550 m/z mass range in the Orbitrap with 120K resolution, while MS/MS data were acquired in DDA mode in the ion trap in top-speed mode (3″ long cycle). Raw data were analyzed by Proteome Discoverer 2.4 (Thermo Fisher Scientific) using the Homo Sapiens database, downloaded from UniProtKB/Swiss-Prot database (v2017-10-25), with 15 ppm and 0.6 Da tolerance for MS and MS/MS data, respectively. Quantification was based on summed abundance deriving from precursor ion intensity of unique and razor peptides. Significant ratios were calculated as pairwise ratios using *t*-test (background based) function embedded in Proteome Discoverer. Maximum allowed fold change was set to 100 in the absence of a detected abundance sample group. Protein ratios were considered significant with fold change > 2, Benjamini-corrected *p*-value < 0.05 and with at least three of four detected abundance values in at least one sample group. All the mass spectrometry data have been deposited to the ProteomeXchange Consortium via the MassIVE partner repository with the MSV000092172 dataset identifier.

### 2.11. Immunocytochemistry

Immunofluorescence stain of 3D PNI assays was made incubating the system with 10% normal donkey serum, 0.3% Triton X-100 as blocking solution overnight (ON) at 4 °C. The following day, primary anti-GFAP antibody (code Z0334, Dako, Agilent Tecnologies; 1:50 in blocking solution) was incubated ON at 4 °C. After three 5′ PBS washes, secondary antibody Cy3, anti-rabbit IgG, diluted 1:400 was incubated ON at 4°C. After three 5′ PBS washes, the glass slides were mounted with Vectashield onto microscope slides, and micrographs were taken under Eclipse E600 (Nikon Instruments S.p.A) fluorescence microscope. For all immunoreactions, negative controls (primary antibody replaced by blocking solution) were also included.

### 2.12. Statistical Analyses

All in vitro experiments consisted of at least three biological replicates. Statistical analyses were conducted using GraphPad Prism version 9.00 software. Comparisons were made using Student’s *t*-test, Mann–Whitney U-test, Fisher’s exact test and one-way ANOVA, when appropriate. Differences were considered significant for *p* values ˂ 0.05.

## 3. Results

### 3.1. Nerve-Infiltrating CCA Cells Show Lower Cytokeratin 7 Positivity Compared to the Tumor Mass in Human Intrahepatic CCA Samples

#### 3.1.1. Patient Characteristics and Sample Selection

To look for any phenotypic difference induced by perineural invasion (PNI) in nerve-infiltrating CCA cells with respect to the main tumor mass, we collected samples from N = 33 iCCA patients who underwent liver resection with curative intent.

To identify samples with PNI, the histological assessment of tissues was performed on H&E-stained slides. Upon examination, N = 6/33 iCCA showed PNI within the section (Figure 1A). The rate of PNI in iCCA observed in our samples agrees with the medical literature [23,24]. We selected samples with PNI to perform further histomorphological analyses. Patient characteristics in terms of age, gender and tumor staging are reported in Table 1.

#### 3.1.2. In Situ Phenotype of Tumor Cells Invading Nerves

To individuate the tumor phenotypical traits associated with PNI, positivity for the mature biliary epithelial marker cytokeratin 7 (CK7) was assessed in the tumor mass and in tumor cells invading the nerves (Figure 1B). Interestingly, our data show that in iCCA samples, tumor cells invading nerves were characterized by a lower positivity for CK7 compared to the tumor mass (15.1 ± 12.4% vs. 48.3 ± 14.6%; *p* < 0.01). To understand more about how interaction with nerves could induce this different phenotype in cancer cells of the iCCA group, we decided to study in vitro the interaction between iCCA-derived cell lines and peripheral nerve explants.

### 3.2. HuCC-T1 Cells Migrate towards Sciatic Nerve Explants through an Extracellular Matrix Scaffold in a 3D Model of Perineural Invasion

To analyze interactions between iCCA cells and peripheral nerves, we simulated perineural invasion in vitro following the protocol described by Demir and colleagues [7]. Figure 2 illustrates the experimental system, consisting of two extracellular matrix (ECM) gel drops connected by an ECM gel bridge. The drop on the left contained a sciatic nerve explant; the one on the right contained a suspension of HuCC-T1 cells (a cell line derived from a human iCCA). One day after plating, the nerve explant started its degeneration (panel A), while HuCC-T1 cells remained at the original plating site (panel B). Seven days after plating, HuCC-T1 cells consistently migrated from the original plating site towards the boundary of the ECM gel drop (panel D) containing the degenerated nerve explant (panel C). Interestingly, chains of GFAP+ activated Schwann cells (SCs) from the nerve reached out towards cancer cells (panel E). Based on these early data, we could assume that the peripheral nerve, and presumably SCs, produced soluble factors capable of promoting the migration of iCCA cells.

### 3.3. Human SC-Conditioned Medium Increases the Migration of iCCA-Derived Cells

Thus, to evaluate the contribution of soluble factors released by SCs in favoring HuCC-T1 neurotropism, we tested the ability of the conditioned medium (CM) of human Schwann cell primary cultures (hSCs) to induce the migration of HuCC-T1 across a transwell. The control consisted of medium conditioned by HuCC-T1 cells themselves (hereafter control CM). We evaluated the number of migrated cells and concluded that hSCsCM significantly increased the migration of HuCC-T1 cells compared to the control (Figure 3A). These data were replicated in another iCCA-derived cell line, namely Oz [25], obtaining almost overlapping results (Figure 3B). Taken together, these data suggest that hSCs release secreted factors in the conditioned media that have a chemotactic effect on iCCA-derived cells.

### 3.4. Human SC-Conditioned Medium Increases Invasion and Induces Cadherin Switch in HuCC-T1 Cells

Next, we decided to evaluate whether, in addition to migration, hSC CM could also affect tumor invasiveness. In line with the preceding data, we found a greater propensity of HuCC-T1 cells to invade Matrigel when exposed to hSC CM with respect to control CM (Figure 4). In the same experimental conditions, Oz cells displayed a trend (yet statistically unsignificant) of increased Matrigel invasiveness (Appendix A). To test whether HuCC-T1’s observed enhanced invasiveness was associated to an epithelial–mesenchimal transition (EMT), we performed Western blot analyses comparing single cultured HuCC-T1 with HuCC-T1/hSCs indirect cocultures. Exposure to hSC-secreted factors determined a cadherin switch in HuCC-T1 cells, consisting of the downregulation of the epithelial marker E-cadherin, accompanied by the coherent upregulation of the mesenchymal marker N-cadherin (Figure 5). An upregulation trend (yet statistically unsignificant) of the mesenchymal marker N-cadherin was found in Oz/hSCs indirect coculture, as compared to single cultured Oz cells (Appendix A).

Since low CK7 immunohistochemistry positivity was detected in tumor cells invading nerves (Figure 1B), we performed Western blot analysis of CK7 in HuCC-T1 cells treated with hSC CM. The results reported in Appendix A show a decrease in CK7 protein in HuCC-T1 cells exposed to hSC CM with respect to control CM.

### 3.5. Human SC-Conditioned Medium Stimulates Cell Viability and Proliferation of iCCA Cells

#### 3.5.1. Human SC-Conditioned Medium Increases HuCC-T1 Cell Viability

To investigate whether hSCsCM treatment could affect cell viability, we plated HuCC-T1 cells at two different cell densities (see Materials and Methods). These cells were treated with hSCsCM or control CM for 48 h. With either cell densities, a significant increase in cell viability was observed in samples exposed to hSC CM, as compared to control CM (Figure 6). Notably, the effect of hSC CM treatment remains as pronounced, even when seeding CCA cells at a suboptimal cell density. The pro-survival effect of the hSC CM, observed also at suboptimal cell density in iCCA cells, prompted us to perform a clonogenic assay.

#### 3.5.2. Human SC CM Increased Clonogenicity of HuCC-T1 Cells

We treated HuCC-T1 cells with either hSC CM or control CM to evaluate the number of colonies formed in the two experimental groups. We observed a statistically significant increase in the number of colonies formed by CCA cells treated with hSC CM, as compared to control CM, confirming a pro-survival effect of hSC CM (Figure 7, left). Moreover, by monitoring the morphology of the colonies, we noticed that in hSC CM-treated samples, they appeared bigger with respect to those formed in control CM. This observation seems to suggest the occurrunce of an intense proliferation in hSC CM-treated colonies. In addition, some of the hSC CM-treated colonies were composed of spindle-like cells who lost cell–cell contact and appeared scattered. This finding is in line with the hypothesis that hSC CM induces EMT in HuCC-T1 cells (Figure 7, right).

#### 3.5.3. Human SC CM Increases HuCC-T1 Proliferation

Lastly, we evaluated proliferation by BrdU incorporation in HuCC-T1 treated with hSC CM or control CM. Consistent with previous cell viability and clonogenicity data, we observed an increased proliferation after 24 and 48 h in samples treated with hSCsCM, as compared to the controls (Figure 8). In addition, proliferation was assessed in Oz at 48 h of hSC CM exposure, considering that these cells have a longer doubling time with respect to HuCC-T1. Remarkably, as already observed in HuCC-T1, in Oz cells, we also found increased BrdU incorporation following treatment with hSC CM with respect to control CM (Figure 8). The cell viability, clonogenic and BrdU assays, altogether, consistently indicate that hSC CM has a proliferative and pro-survival effect on iCCA cells.

### 3.6. Differential Proteome Analysis of HuCC-T1 Cells Cocultured with Human SCs

To gain insight into the molecular mechanism underlying the observed phenomena, we compared HuCC-T1 single cultured or cocultured with primary hSCs (as detailed in the Methods section). We analyzed HuCC-T1 single culture versus the HuCC-T1/hSCs coculture proteome using mass spectrometry, quantifying, in total, 4938 proteins with at least 2 peptides and identifying 10 upregulated and 10 downregulated proteins. Interestingly, we found six proteins related to EMT, migration and invasion and six proteins related to proliferation and survival, with four proteins involved in both effects (Table 2). Remarkably, we quantified collagen VII alpha-1 chain only in coculture samples, being undetectable in controls. It is of interest that in other cellular models, TGFβ is a key promoter of collagen VII gene expression [26,27], leading us to hypothesize its possible role in mediating at least some of the effects determined by the hSCs exposure in our iCCA cultures. A complete list of both downregulated and upregulated proteins with the respective accession numbers and *p*-values in HuCC-T1 single cultures versus HuCC-T1/hSCs cocultures is provided in Appendix A.

### 3.7. The Effect of hSC CM on the Migration and Invasion of iCCA Cells Is Blocked by an Antagonist of TGFBR1

To test the hypothesis that hSC-derived TGFβ was a possible inducer of migration and invasion in iCCA cells, we carried out transwell migration and invasion experiments by treating HuCC-T1 or Oz cells in the presence of a specific TGFβ receptor 1 (TGFBR1) antagonist, namely SB-431542 (10 μM). As shown in Figure 9, the incubation with SB-431542 completely reverted the effect of hSC CM, both in HuCC-T1 and Oz cells, only slightly affecting the controls (Figure 9). These data suggest the pivotal role of TGFβ in regulating the migration/invasion of iCCA cells.

## 4. Discussion

In recent years, the role of nerves in tumor progression has been clarified. While different articles elucidated the contribution of neurotransmitters and nerve fibers to cancer growth [3,4], little is known about the role of glial cells. Recently, Gundlach and colleagues documented the presence of SC markers in human CCA sections and their association with a worse prognosis [14]. In the present study, we evaluated the ability of such glial cells to promote CCA progression. Analyzing sections of iCCA from patients, we observed a lower positivity of CK7 in nerve-infiltrating cancer cells compared to the cells in the tumor mass. This finding was confirmed in our in vitro model via Western blot analysis. The altered expression of CK7, being a biliary epithelial marker, could possibly imply an EMT. Therefore, we decided to further investigate in vitro the interactions between the iCCA cell line HuCC-T1 and a nerve explant through a 3D PNI assay. We found migration of cancer cells towards the nerve explant, reaching GFAP+ SCs coming from the nerve. It has been previously reported in pancreatic ductal adenocarcinoma, another model of cancer–glia interaction, that chains of GFAP+ SCs are implied in triggering PNI by actively recruiting cancer cells to transport them on nerve fibers. Interestingly, these SCs line up, in a tip-to-tail manner, to form chains resembling Büngner bands [11,12], a feature we also observed in our 3D PNI assay. In human iCCA-derived cell lines (HuCC-T1 and Oz cells), migration appears to be positively regulated by hSCs, as assessed by our transwell migration assays. Remarkably, Oz cell migration towards hSCsCM was higher than HuCC-T1. This may be due to the more aggressive behavior of the primitive CCA, from which the cell line was derived. In fact, Oz cells are regarded as rapidly metastasizing, while HuCC-T1 is only moderately invasive in relation to the different degree of invasiveness of the primary tumors [25]. Moreover, we also documented increased invasiveness through Matrigel of HuCC-T1 cells induced by hSC CM. For internal consistency, we evaluated Oz invasiveness at the same time point and found a non-significant trend of increasing invasion of Oz cells treated with hSC CM, as compared to control CM. Due to the lower expression of CK7 in nerve-infiltrating iCCA cells observed on sections and the increased invasiveness of HuCC-T1 cells exposed to hSC CM in vitro, we tested the hypothesis that cancer cells underwent epithelial–mesenchymal transition when exposed to secreted factors produced by hSCs. Comparing HuCC-T1 lysates obtained from either single cultures or HuCC-T1/hSC indirect cocultures via Western blot, we found evidence of a cadherin switch (downregulation of E-cadherin with concomitant upregulation of N-cadherin). This is a feature commonly observed in TGFβ-induced epithelial–mesenchymal transition. It is remarkable that a cadherin switch alone correlates with reduced survival rates in patients [34]. Performing cell viability, clonogenic and proliferation assays on HuCC-T1 and Oz cells exposed to hSC CM, we also revealed a proliferative and pro-survival effect of hSCs.

To gain insights into a potential mechanism for SC-driven CCA progression, we performed a differential proteome analysis on HuCC-T1 lysates from single cultures or indirect cocultures with hSCs. Interestingly, we documented the selective modulation of only twenty proteins, ten upregulated and ten downregulated. We observed the upregulation of key oncogenes described in other tumoral models, such as PN1 [28] and NDUFC1 [30], while, on the other hand, we found downregulation of the crucial tumor suppressors PP2A [29,31,35], ENO3 [32] and NOTCH3 [33]. Among the modulated proteins, collagen VII was detected only in HuCC-T1 lysates from cocultures, suggesting the modulation of TGFβ signaling. In fact, as reported in other cellular models, TGFβ promotes gene expression of collagen VII [26,27]. Considering all the evidence gathered (cadherin switch, migration, invasion, collagen VII upregulation), we decided to test the role of TGFβ in our system. To this end, the migration and invasion assays were performed in the presence of the specific TGFBR1 antagonist SB-431542, obtaining the almost complete reversion of the hSC CM-induced effects.

We are, to our knowledge, the first to demonstrate enhanced malignant traits (migration, invasion, proliferation and survival) in iCCA cells, induced by SCs, which appear to be TGFβ-dependent. Our contribution may be a starting point for further investigations concerning the mechanism of such enhancements. In fact, our proteomic data offer interesting hints to better characterize iCCA/SCs crosstalk. PP2A is a master negative regulator of cell proliferation (via Wnt/β-catenin and AKT) and survival (*via* BAX and BCL2, promoting apoptosis). It also inhibits EMT by dephosphorylating β-catenin and allowing its degradation to occur [29]. In CCA, and more specifically HuCC-T1 cells, the role of PP2A as an inhibitor of EMT, invasiveness and proliferation has been clarified [31,35]. Notably, PP2A inhibits TGFβ signaling by TGFBR1 dephosphorylation, and downregulation of PP2A in prostate cancer leads to enhanced TGFβ signaling [36]. Our proteomic analysis reveals no modulation of SET and CIP2A, the endogenous regulators of PP2A (data not shown). Interestingly, there is evidence of calcium-mediated regulation of PP2A via CRE [37], controlling the expression of its catalytic subunit (PP2Ac), which we found downregulated in our model. It appears promising to further characterize the SC-induced downregulation of PP2Ac as it could facilitate TGFβ signaling and directly affect proliferation, EMT, invasion and survival. We look forward to being able to better clarify the role of the downregulations of ENO3, which converge on the Wnt/β-catenin pathway [32] and NOTCH3, as it upregulates p21, blocking the cell cycle [33]. The use of neutralizing antibodies against TGFβ has been tested in mice and humans (fresolimumab), and systemic side effects are reversible or self-limiting [38]. Considering our TGFBR1 antagonist experiments, we believe it could be of interest to test anti-TGFβ antibodies in our system as it could open new potential therapeutic approaches.

## 5. Conclusions

Taken together, our results indicate that Schwann cells assist in promoting greater aggressiveness in cholangiocarcinoma (CCA). The data reported here refer only to intrahepatic CCA, and we aim to extend our analysis to extrahepatic CCA in a subsequent study. Overall, it is generally recognized that in CCA patients, the occurrence of perineural invasion (PNI) is associated with a worse prognosis. Despite the accepted relevance, the molecular mechanism underlying this association currently remains elusive. In human CCA specimens, we detected a lower positivity for the epithelial marker cytokeratin 7 (CK7) in nerve-infiltrating CCA cells with respect to the tumor mass. Concurrently, in CCA cell lines exposed to the secreted factors of human Schwann cell primary cultures, we observed increased migration/invasion and survival/proliferation together with the upregulation of key oncogenes and downregulation of tumor suppressors, as determined by proteomic analysis. It emerges that many of the regulated proteins are under the control of TGFβ, and a receptor antagonist of the TGFβ receptor blocks the Schwann-cell-induced migration/invasion of CCA cells.

To our knowledge, this is the first evidence that Schwann cells promote malignant traits in CCA, opening new perspectives for the interpretation of PNI in this tumor type.

## Figures and Tables

**Figure 1 cells-13-00366-f001:**
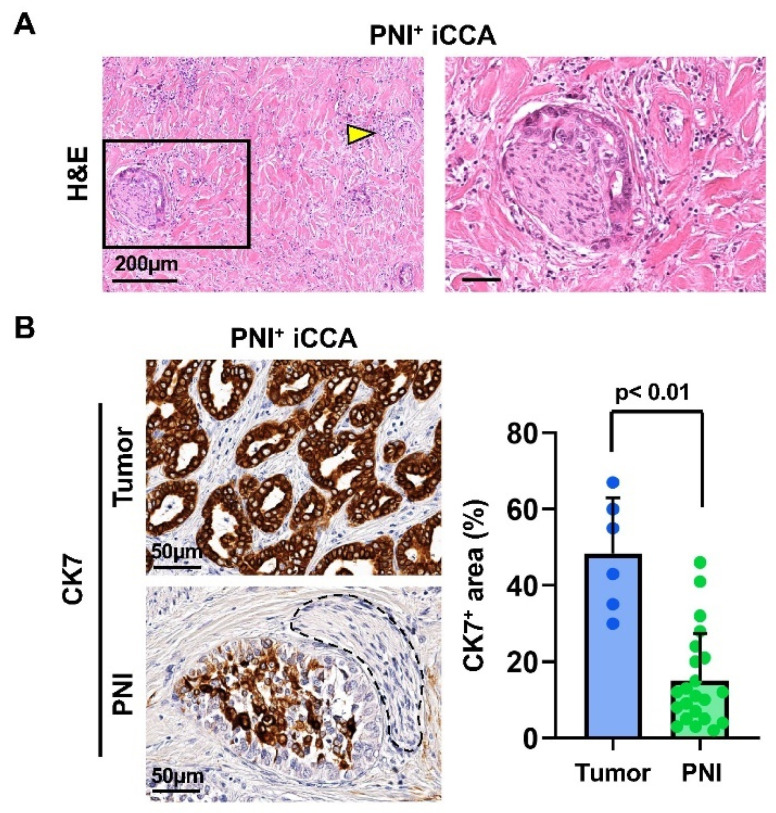
In situ histomorphological and immunohistochemical study of nerve phenotype in intrahepatic cholangiocarcinoma (iCCA) samples with perineural invasion (PNI+). (**A**). Hematoxylin and eosin (H&E) stain. Nerves without infiltration (spared) and those infiltrated were present within the same tissue slide. Yellow arrowhead indicates a spared nerve, while the area in the box contains a nerve with PNI, magnified in the right panel. (**B**). Immunohistochemistry for cytokeratin 7 (CK7) in CCA tumor mass (upper panel) vs. PNI (lower panel). Dotted lines individuate an infiltrated nerve. Histogram shows means and standard deviation of the CK7+ cell area percentage. Statistical significance was assessed with Mann–Whitney U-test.

**Figure 2 cells-13-00366-f002:**
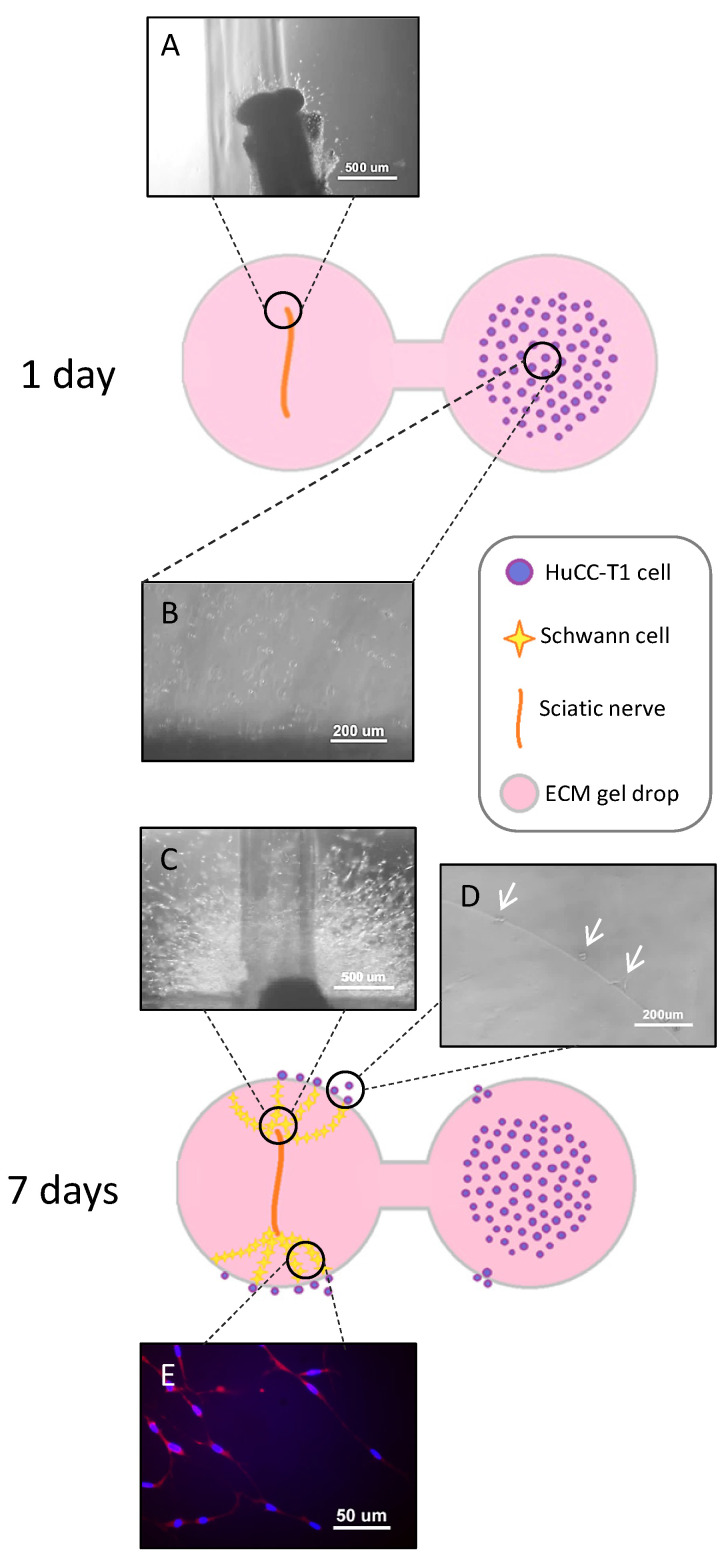
Three-dimensional perineural invasion assay. The cartoon shows the experimental system at the beginning (day 1) and the end (day 7) of the experiment. On the right, the components of the system are listed. The micrographs (**A**–**E**) show details of the experiment. (**A**) incipient degeneration of sciatic nerve (SN); (**B**) HuCC-T1 original plating site; (**C**) degenerated nerve surrounded by cells emerging from it; (**D**) HuCC-T1 cells (white arrows) approaching the gel drop containing the nerve explant; (**E**) immunoflorescence staining for GFAP (red) shows activated SCs chains. Nuclei are labelled by DAPI (blue). N = 5.

**Figure 3 cells-13-00366-f003:**
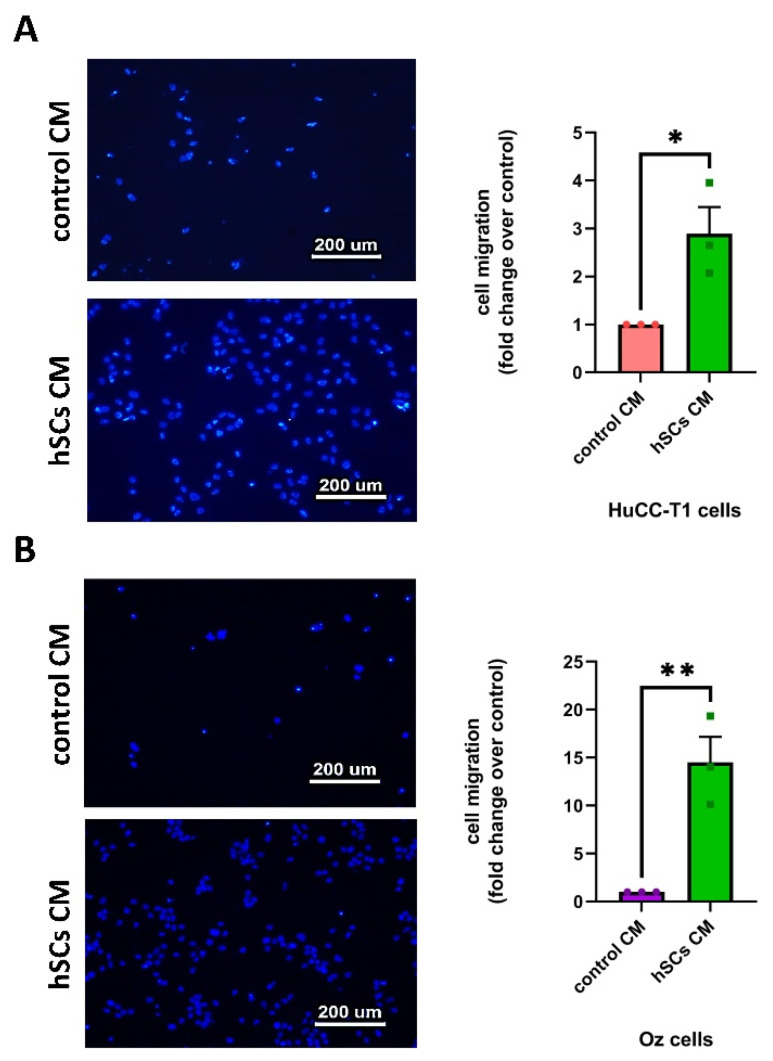
Migration of HuCC-T1 and Oz cells induced by hSC-conditioned medium. Human cell lines derived from iCCA tumors were seeded in the upper chamber of a transwell. Conditioned medium obtained from hSC primary cultures was placed in the lower chamber and migration assessed for HuCCT-T1 cells (panel **A**) and for Oz cells (panel **B**). As shown by the micrographs and related diagrams, hSC-conditioned medium increased the migration of iCCA cells through the insert. Student’s *t*-test * *p* ≤ 0.05; ** *p* ≤ 0.01. N = 3.

**Figure 4 cells-13-00366-f004:**
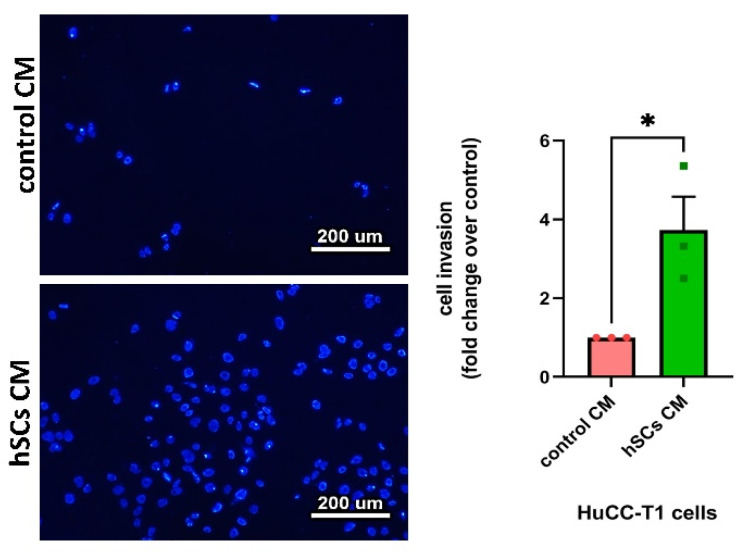
Human SC-conditioned medium fosters the invasiveness of HuCC-T1 cells in a Matrigel invasion assay. Micrographs and related diagrams show the higher propensity of HuCC-T1 cells to invade Matrigel when exposed to hSC CM with respect to control CM. Student’s *t*-test: * = *p* ≤ 0.05. N = 3.

**Figure 5 cells-13-00366-f005:**
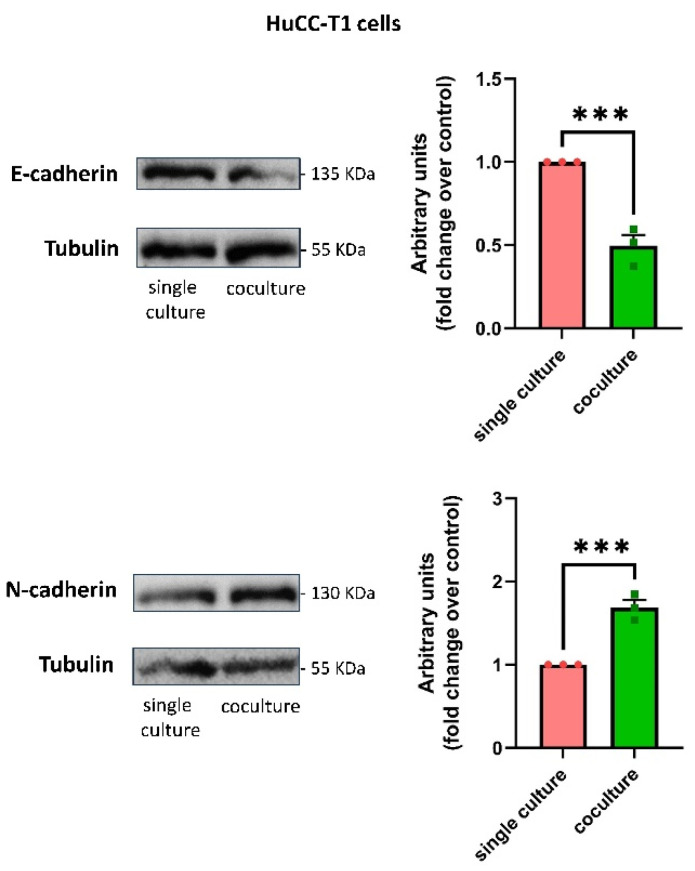
Cadherin switch in HuCC-T1 cells cocultured with hSCs. Western blot analyses of E-cadherin and N-cadherin expression in HuCC-T1 cells cocultured with hSCs. The figure shows the downregulation of the epithelial marker E-cadherin and the coherent upregulation of the mesenchymal marker N-cadherin. The graphs depict the results of densitometric analyses. Student’s *t*-test: *** *p* ≤ 0.001. N = 3.

**Figure 6 cells-13-00366-f006:**
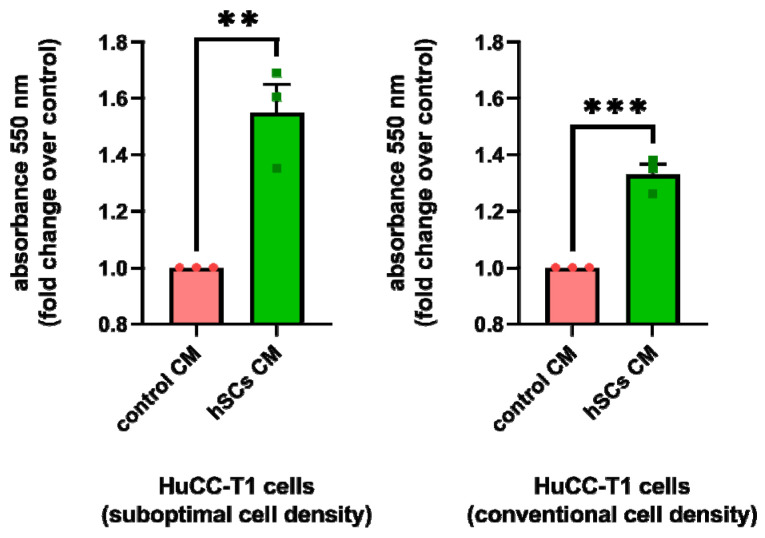
Cell viability assay in hSC CM-treated HuCC-T1 cells. The graphs show viability according to MTT assay in HuCC-T1 cells treated with hSC CM or control CM. Cell viability is higher in samples treated with hSC CM with respect to control at both conventional and suboptimal cell densities. Student’s *t*-test: ** *p* ≤ 0.01; *** *p* ≤ 0.001. N = 3.

**Figure 7 cells-13-00366-f007:**
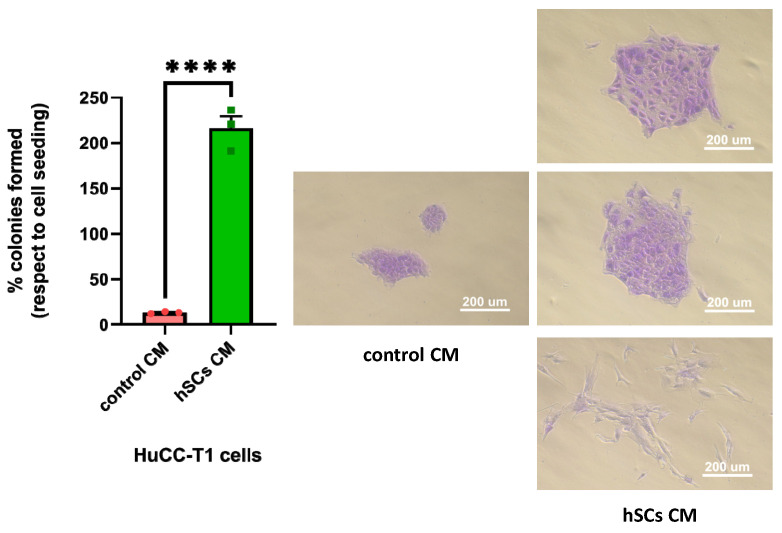
Clonogenic assay in hSC CM-treated HuCC-T1 cells. The graph shows higher percentage of colonies formed with respect to initial seeding in HuCC-T1 cells treated with hSC CM compared to the control. Student’s *t*-test: **** *p* ≤ 0.0001. N = 3. The micrographs show the different morphology of the colonies formed by cells treated with hSC CM and control CM.

**Figure 8 cells-13-00366-f008:**
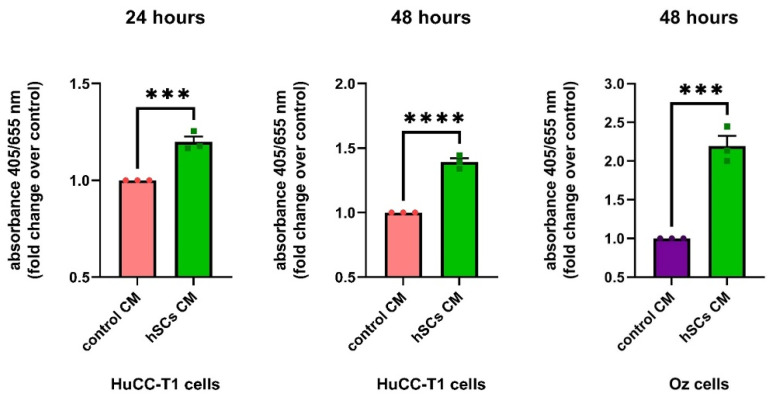
Proliferation assay in hSC CM-treated HuCC-T1 and Oz cells. The graphs show increased proliferation of both cell lines treated with hSC CM compared to control CM. Student’s *t*-test: *** *p* ≤ 0.001 **** *p* ≤ 0.0001. N = 3.

**Figure 9 cells-13-00366-f009:**
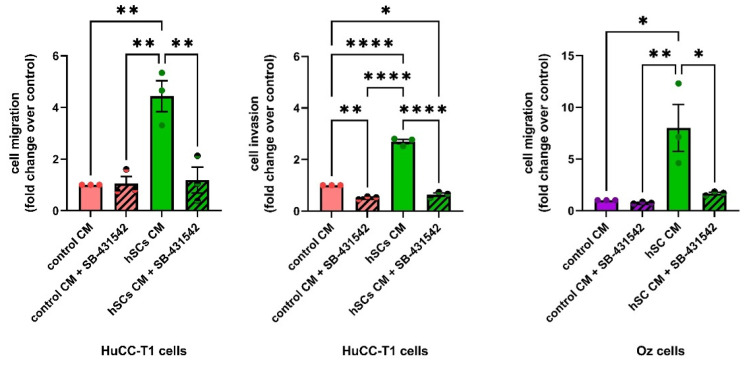
Migration and invasion assays in the presence of TGFBR1 antagonist treatment. Migration and invasion assays were performed in iCCA cells treating HuCC-T1 or Oz cells with the TGFBR1 antagonist (10 μM SB-431542). The graphs show that the antagonist completely reverts the effects of hSC CM. One-way ANOVA * *p* ≤ 0.05; ** *p* ≤ 0.01; **** *p* ≤ 0.0001. N = 3.

**Table 1 cells-13-00366-t001:** Patient characteristics and tumor staging. MF: mass forming; PI: periductal infiltrating; IG: intraductal growth; TNM: tumor (T), nodes (N), metastases (M). For age, data are expressed as median (IQR).

	iCCA with PNI
Age (years)	70 [64–73]
Gender	F: 3
	M: 3
Growth pattern	MF: 5
	PI: 1
	IG: 0
TNM	
T	T1: 0
	T2: 3
	T3: 3
	T4: 0
N	N0: 5
	N1: 1
M	M0: 6
	M1: 0
Stage	I: 0
	II: 3
	III: 3
	IV: 0

**Table 2 cells-13-00366-t002:** Proteins differentially regulated in HuCC-T1 cells cocultured with hSCs identified by proteomic analysis. The table summarizes the most relevant modulated proteins in HuCC-T1/hSC coculture with respect to controls. Red: upregulated proteins; blue: downregulated proteins. *t*-test, Benjamini-corrected *p* < 0.05. N = 4.

Protein Name	Involvement in Migration, Invasion, EMT (TGFβ Pathway)	Involvement in Proliferation, Survival	Regulation
Collagen alpha-1 (VII) chain [26,27]	X		↑
Protease Nexin 1 (PN1) [28,29]	X		↑
NADH dehydrogenase [ubiquinone] 1 alpha subcomplex subunit 1 (NDUFC1) [30]	X	X	↑
Cytoplasmic dynein 1 intermediate chain 1		X	↑
Protein 4.1		X	↑

Serine/threonine-protein phosphatase 2A catalytic subunit alpha isoform (PP2Ac) [29,31]	X	X	↓
Beta-Enolase 3 [32]	X	X	↓
Neurogenic locus notch homolog protein 3 (NOTCH3) [33]	X	X	↓

## Data Availability

The data presented in this study are openly available in MassIVE repository with the MSV000092172 dataset identifier.

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
