# Peer review of "Cholangiocarcinoma Malignant Traits Are Promoted by Schwann Cells through TGFβ Signaling in a Model of Perineural Invasion"

_cells, 2024, doi:10.3390/cells13050366_

Round 1
Reviewer 1 Report
Comments and Suggestions for Authors
De Franchis et al studied the interaction between Schwann cells (SC) and cholangiocarcinoma (CCA) cells, characterizing how SC may promote CCA malignant traits. The authors observed differences in CK7+ area in patients with iCCA with perineural invasion (PNI), when compared with the rest of the tumor mass. Importantly, this was observed only in iCCA tumours, while no differences were observed in eCCAs. Using an in vitro 3D model of CCA cells and SCs, the authors observed that CCA cells move towards the peripheral nerve explant and get into contact with SC. Conditional media (CM) from SC was shown to increase CCA cell proliferation, migration and colony formation. They observed that TGFb might be an important mediator in this process. Although this work contains important and relevant findings in the field, there are several important experiments missing to confirm the observations and confirm the interaction between SC and CCA cells. Therefore, several additional experiments should be conducted.
- The number of samples included in the study is limited and therefore it is not possible to perform prognostic analysis. Since the authors observe some differences between iCCA and eCCA patients, it would be important to also discuss if there are differences in the prognosis of patients with iCCA/eCCA and PNI.
- In Fig.1, the authors reported that a lower expression of CK7 in nerve-infiltrating tumor cells is observed in iCCA samples, when compared with the tumor mass. This conclusion is not supported by the data. The authors quantified the % of CK7+-cells and this type of experimental approach (IHC) is not the most suitable to conclude about differences in expression levels, but only about presence/absence. The differences in the area observed might result from differences in infiltration rather than expression of CK7. Please revise it carefully and also measure the expression of CK19 and SOX9.
- An important comment for all the story is that the authors should compare the findings obtained with iCCA cells with another experiments using eCCA cells, to observe if the observed findings are specifically detected in iCCA or a more general event.
- In addition to conducting experiments with conditional media, it would be important to also conduct co-culture experiments and measure the same phenotypical outcomes.
- Some wound healing assays would also be of value to conclude about the impact of SC in CCA cell invasion.
- Regarding the proteomic analysis, the strategy followed might not be the best one to draw relevant conclusions. Instead of evaluating the proteome of SC-CCA co-cultures, the authors should: 1) evaluate the secretome of SC in order to try to identify the potential mediators; 2) evaluate the proteome of CCA cells previously exposed to SC conditional media. In addition, a detailed proteomic analysis should be presented, including, heatmaps, gene ontology and geneset enrichment analysis in order to conclude about the deregulated processes
- Regarding the experiment with TGFb inhibitor, the most important experiment should be treating SC cells with this inhibitor and then use the conditional media to treat CCA cells. With the current experimental layout, it is difficult to take conclusions.
Author Response
We thank the Reviewer for the useful criticisms; we addressed all issues raised by the Reviewer.
Question n. 1: The number of samples included in the study is limited and therefore it is not possible to perform prognostic analysis. Since the authors observe some differences between iCCA and eCCA patients, it would be important to also discuss if there are differences in the prognosis of patients with iCCA/eCCA and PNI.
Answer: We thank the reviewer for this comment; we plan to deeply investigate eCCA in a future study. To enhance the cohesiveness of the manuscript, we took the decision to narrow our attention to iCCA samples only. Focusing on iCCA samples also complies with a specific suggestion from Reviewer n. 3. Therefore, in the revised manuscript, eCCA in vivo results have been removed from Figure 1, but, in order to comply with the Reviewer n. 1 specific request, the numerosity of iCCA patients was increased from 20 up to 33. Paragraph 3.1, Fig.1 and Table 1 have been changed accordingly. In the revised form of the manuscript we added a sentence in the Introduction to indicate the association of PNI with CCA worse prognosis (lines 39-41, page 1).
Question n. 2: In Fig.1, the authors reported that a lower expression of CK7 in nerve-infiltrating tumor cells is observed in iCCA samples, when compared with the tumor mass. This conclusion is not supported by the data. The authors quantified the % of CK7+-cells and this type of experimental approach (IHC) is not the most suitable to conclude about differences in expression levels, but only about presence/absence. The differences in the area observed might result from differences in infiltration rather than expression of CK7. Please revise it carefully and also measure the expression of CK19 and SOX9
Answer: We thank the Reviewer for this comment. In the revised manuscript, we replaced “expression of CK7” with the more correct “positivity for CK7” (pages 2 , 3, 6, 7 and 15). The positivity was evaluated as area % of the tumour cells infiltrating the nerve, not of the whole nerve. Therefore, the counting was not affected by the extent of nerve infiltration and it reflects the actual percentage of the tumour cells positive for CK7.
Also, to corroborate IHC findings and to give greater consistency and coherency to the data and to meet the reviewers' requests, we evaluated whether exposure of HuCC-T1 cells to the conditioned medium of hSCs was sufficient to drive a decrease in CK7 protein expression. Results of the western blot analysis of HuCC-T1 cells cultured with hSCs CM are displayed in the Supplementary Fig 3 (Fig. S3) showing a significant down regulation of CK7 expression following hSCs CM exposure.
The following sentence has been added to paragraph 3.4:“Since low CK7 immunohistochemistry positivity was detected in tumor cells invading nerves (Figure 1B), we performed Western blot analysis of CK7. The results reported in Fig. S3 show a decrease of CK7 protein in HuCC-T1 cells exposed to hSCs CM respect to control CM." (lines 350-353, page 10). Furthermore, to support the relevance of the observed quantification, the following sentence has been added to the Discussion: “This finding was confirmed in Western blot analyses” (line 477 page 15).
We opted to not explore CK19 and Sox9 positivity. We agree that CK19 and Sox9 are considered as markers of biliary epithelial cells; however, both antigens are not reliable markers for CCA cells with variable expression. In particular, CK19 shows an overlap with CK7 (doi.org/10.1002/hep.20130), thus not furnishing additional information; also, CK19 positivity in cholangiocarcinoma is less consistent compared to CK7 (doi.org/10.1016/j.mpdhp.2024.01.004). Regarding Sox9, it has a variable expression in CCA (doi.org/10.1038/s41416-018-0338-9) and is also ubiquitously expressed in several tissues/cells including glial cells (nerves); therefore in a separate specific study we are currently investigating additional markers showing high specificity and sensibility.
Question n. 3: An important comment for all the story is that the authors should compare the findings obtained with iCCA cells with another experiments using eCCA cells, to observe if the observed findings are specifically detected in iCCA or a more general event.
Answer: We thank the reviewer for this comment. As mentioned in the answer to question n. 1, we plan to deeply investigate eCCA in a future study. All the experiments on eCCA cell lines would require additional reagents that would take additional months, exceeding the special issue deadline. Trying to comply with the Reviewer’s request as much as possible, we refocused the revised paper on intrahepatic CCA subtype only, to increase the consistency within the entire manuscript. This also complies with a specific suggestion from Reviewer n. 3.
Question n. 4: In addition to conducting experiments with conditional media, it would be important to also conduct co-culture experiments and measure the same phenotypical outcomes.
Answer: We thank the reviewer for this comment. According to the reviewer comment, the experiments depicted in Figure 5 and in the Supplementary Fig. S2, as well as proteomic analyses, were performed testing cell-lysates obtained from co-cultures experiments.
Question n.5: Some wound healing assays would also be of value to conclude about the impact of SC in CCA cell invasion.
Answer: We thank the reviewer for this comment. We selected Transwell assays to investigate invasion. In this model, cells have to digest Matrigel first and then go through 8μm pores in order to pass across the transwell insert and reach the other compartment. On the other hand, in wound healing assay, since cells do not have to move through a porous membrane, a proliferation bias could interfere with the interpretation of the results. Therefore, we believe migration/invasion assays reported in the current study in Figures 3, 4, 9 and Supplementary S1 can be considered assays testing wound healing properties.
Question n. 6: Regarding the proteomic analysis, the strategy followed might not be the best one to draw relevant conclusions. Instead of evaluating the proteome of SC-CCA co-cultures, the authors should: 1) evaluate the secretome of SC in order to try to identify the potential mediators; 2) evaluate the proteome of CCA cells previously exposed to SC conditional media. In addition, a detailed proteomic analysis should be presented, including, heatmaps, gene ontology and geneset enrichment analysis in order to conclude about the deregulated processes.
Answer: We thank the reviewer for this comment and for giving us the opportunity to better explain our experimental strategy. The purpose of the proteomics experiments was to clarify which factors produced by hSCs might be responsible for the increased proliferation/migration/invasion of iCCA cells. Other authors have previously published the proteomic analysis of the secretome of hSCs under basal conditions, and therefore we did not repeat these experiments. Specifically, Ferdoushi and colleagues (Frontiers in Oncology, 2020) performed a proteomic analysis of the secretome of hSCs obtained from the same source of ours (Sciencell).
We understand the reviewer expectation about the graphical pictures and enrichment analyses that usually describe a proteomic analysis, but in this case the proteomic analysis found only 20 proteins (shown in Tab. S1) statistically up- or down-regulated and these don't represent any enriched pathway (even if we looked for it). Some of the modulated proteins (and among them collagen VII) are under the control of TGFβ (Tab.1) and this fact led us to consider TGFβ as a possible mediator of the observed biological effects. This hypothesis was confirmed by the experiments conducted with a TGFβ receptor antagonist in both the iCCA cell lines investigated in this study.
Question n.7: Regarding the experiment with TGFb inhibitor, the most important experiment should be treating SC cells with this inhibitor and then use the conditional media to treat CCA cells. With the current experimental layout, it is difficult to take conclusions.
Answer: We thank the reviewer for this comment. SB-431542 is an antagonist of TGFβ receptor I, which prevents phosphorylation of downstream effector proteins within target cells. Therefore, TGFBR1 antagonist SB-431542 recognizes its target on cancer cells, while SC cells produce the corresponding ligand TGFβ. This explains why the migration/invasion experiments depicted in Figure 9 were performed using CCA cells exposed to the CM in the presence of the inhibitor.
By treating cancer cells with TGFB1R antagonist we switch off any TGFβ-dependent response in cancer cells. The observation that migration/invasion effects induced by hSCs CM are abolished by SB-431542 treatment implies that these effects are TGFβ-dependent.
Reviewer 2 Report
Comments and Suggestions for Authors
The manuscript by de Franchis and colleagues dissect the mechanisms linking the perineural invasion (PNI), a common feature of both intra and extra hepatic cholangiocarcinoma (iCCA and eCCA, respectively), with neoplastic cells aggressiveness. The work is potentially of great interest because it attempts to shed light on a gray area in the pathogenesis of CCA. In fact, it has been known for long time that, in CCA, PNI correlates with a worse outcome of patient, but the molecular mechanisms involved are not known.
The manuscript is clear and well written and the techniques used are adequate. The work includes both morphological analyzes of archival human samples, in vitro experiments and proteomic analysis. Unfortunately, the enthusiasm for this work is diminished by some limitations of the experimental design.
The main weakness of this article is given by the in vitro part. It is not clear why some experiments were in fact conducted using both the iCCA cell lines selected by the authors (HUCCT-1 and Oz), while others only with HUCCT-1. To demonstrate that the results are not cell line-dependent all experiments should be performed with both cell lines. Furthermore, given that PNI is more diffuse among eCCAs than iCCAs (see Results 3.1, line266-267), the experiments could also be conducted on an eCCA line, which would strengthen the significance of the work.
Minor concerns
Table 2: please, check the formatting of the table
Figures 3 and 4: the micrographs are quite dark and difficult to read when printed, please replace them with lighter/brighter pics.
Author Response
We thank the Reviewer for the useful criticisms; we addressed all issues raised by the Reviewer.
Question n.1: To demonstrate that the results are not cell line-dependent all experiments should be performed with both cell lines.
Answer: According to the request, additional experiments have been performed on Oz cells. Oz cells raise technical problems in invasion assays in vitro since they easily aggregate making slower the invasion process. The only study we found in literature addressing Oz cell invasion was performed at 48 hours (Ryu H.S. 2012 Hum. Pathol.). This makes difficult to discriminate between proliferation and invasion effects and justifies why we did not address invasion of Oz cells in the original manuscript.
To comply with the reviewer’s request, we evaluated invasion at 24 hours in Oz cells. As reported in Supplementary Fig 1 (Fig.S1), the invasion assays on Oz cells in the presence of hSCs CM revealed a non-significant increase. The following sentence has been added in paragraph 3.4:“In the same experimental conditions, Oz cells displayed a trend (yet statistically unsignificant) of increased Matrigel invasiveness (Supplementary Fig. S1).”(lines 340-341, page 9)
As the result was not statistically significant, the invasion assay with the specific TGFBR1 antagonist was not performed on Oz cells.
To further comply with the reviewer requests, we also performed N-cadherin Western blot on Oz cells co-cultured with human Schwann cells. A trend increase (however not significant) of the mesenchymal marker N-cadherin was found as compared with single culture. We report such results in the Supplementary Figure 2. The following sentence has been added in paragraph 3.4:“An upregulation trend (yet statistically unsignificant) of the mesenchymal marker N-cadherin was found in Oz/hSCs indirect coculture as compared to single cultured Oz cells (Supplementary Fig. S2).”(lines 347-349, page 10)
Consistently with results obtained in HuCC-T1, also in Oz cells we found that HSCs CM increased BrdU incorporation respect to control. These results are now displayed in the new Figure 8. The following sentence has been added to paragraph 3.5:“In addition, proliferation was assessed in Oz at 48h of hSCs CM exposure considering that these cells have a longer doubling time respect to HuCC-T1.Remarkably,as already observed in HuCC-T1, also in Oz cells we found increased BrdU incorporation following treatment with hSCs CM respect to control CM(Fig. 8).”(lines 400-404, page 12)
Since clonogenic assay requires single-cell suspension for its plating, we could not perform it on Oz cells due to their high tendency to grow in clusters which are extremely difficult to disaggregate. In order to disaggregate Oz cells, trypsinization is not sufficient and several passages through 18 Ga needle are required, which result in extensive cell death (as observed via Trypan blue exclusion test performed during cell counts).
Question n. 2 : Furthermore, given that PNI is more diffuse among eCCAs than iCCAs (see Results 3.1, line266-267), the experiments could also be conducted on an eCCA line, which would strengthen the significance of the work.
Answer: We thank the reviewer for this comment and plan to deeply investigate eCCA in a future study. All the experiments on eCCA cell lines would take additional reagents and several additional months, exceeding the special issue deadline. Trying to comply with the Reviewer’s request as much as possible, we refocused the revised paper on intrahepatic CCA subtypes only, to increase the consistency within the entire manuscript. This also complies with a specific suggestion from Reviewer #3. Therefore, in the revised manuscript eCCA in vivo results have been removed from Figure 1, but the numerosity of iCCA patients was increased from 20 up to 33. Paragraph 3.1,Fig.1 and Table 1have been changed accordingly.
Reviewer #2 minor concerns:
Table 2: please, check the formatting of the table.
Answer: We have checked and modified Table 2.
Figures 3 and 4:the micrographs are quite dark and difficult to read when printed, please replace them with lighter/brighter pics.
Answer: We thank for this comment; Figures 3 and 4 have been improved as requested.
Reviewer 3 Report
Comments and Suggestions for Authors
The authors investigated the role of perineural invasion (PNI) in cholangiocarcinoma (CCA), specifically focusing on its correlation with CK7 expression in human intrahepatic CCA. The study also explored the interaction between an intrahepatic CCA cell line and peripheral nerves in vitro. While the study is intriguing, there are notable areas for improvement.
Major Comments:
- For immunohistochemistry (IHC), it is recommended to increase the number of CCA patients to enhance statistical robustness, especially considering the limited representation of different CCA subtypes. Discrepancies in the reported number of intrahepatic CCA cases between the table and text should be addressed and rectified.
- The authors should consider including extrahepatic CCA cell lines in their in vitro experiments or alternatively refocus the paper on intrahepatic subtypes for consistency in the overall message.
- The rationale behind using OZ cell lines exclusively in the migration assay in Fig. 9 should be clarified. It is suggested to incorporate OZ cell lines as a second intrahepatic CCA cell line in all functional experiments for a more comprehensive analysis.
- Since IHC indicated lower CK7 levels in tumor cells invading nerves, the authors are encouraged to verify this observation by conducting Western blot analysis in HuCCT1 cells cocultured with human Schwann cells (hSCs).
- The list of both downregulated and upregulated proteins in HuCC-T1 single cultures versus HuCC-T1/hSCs cocultures should be provided. Additionally, the absence of p-values in Table 2 should be addressed and rectified.
- In Figure 9, it is recommended to perform the invasion assay for the OZ cell line to enhance the comprehensiveness of the experimental data.
Minor Points:
- The quality of immunofluorescence images in Figures 3 and 4 should be improved, particularly for better visualization of nuclei under different culture conditions.
- The image quality of the Western blot in Figure 5 should be improved for better clarity and accuracy in data representation.
Author Response
We thank the Reviewer for the useful criticisms; we addressed all issues raised by the Reviewer.
Question n.1:For immunohistochemistry (IHC), it is recommended to increase the number of CCA patients to enhance statistical robustness, especially considering the limited representation of different CCA subtypes. Discrepancies in the reported number of intrahepatic CCA cases between the table and text should be addressed and rectified.
Answer: As requested, we have increased the number of CCA patients (from 20 to 33), focusing on iCCA patients only. Table 1 and the text have been accordingly changed. Discrepancies in the reported number of intrahepatic CCA cases between Table 1 and paragraph 3.1 have been addressed and rectified.
Question n.2:The authors should consider including extrahepatic CCA cell lines in their in vitro experiments or alternatively refocus the paper on intrahepatic subtypes for consistency in the overall message.
Answer: As suggested, in the revised manuscript eCCA in vivo results have been removed thus refocusing the paper on intrahepatic subtypes for consistency in the overall message. Paragraph 3.1, Table 1and Fig.1 have been changed accordingly.
Question n. 3a: The rationale behind using Oz cell lines exclusively in the migration assay in Fig. 9 should be clarified.
Answer: We thank the reviewer for this observation. Oz cells raise technical problems in invasion assays in vitro. In fact, they easily aggregate making slower the invasion process. The only study we found in literature addressing Oz cells invasion was performed at 48 hours (Ryu H.S. 2012 Hum. Pathol.). That time length makes it difficult to discriminate between proliferation and invasion effects. We observed a non-significant trend increase of Oz invasion following hSCs CM exposure as compared to control CM at 24 hours (see Supplementary Fig. 1).The following sentence has been added to the Paragraph 3.4: “In the same experimental conditions, Oz cells displayed a trend (yet statistically unsignificant) of increased Matrigel invasiveness (Supplementary Fig. S1).” (lines 340-341 page 9). The following sentence has been added to the Discussion: “For internal consistency, we evaluated Oz invasiveness at the same time point and found a non-significant trend of increasing invasion of Oz cells treated with hSCs CM as compared to control CM.” (lines 493-495 page 15).
Given the non-significant result, the invasion assay with the specific TGFBR1 antagonist was not performed on Oz cells.
Question n.3b: It is suggested to incorporate Oz cell lines as a second intrahepatic CCA cell line in all functional experiments for a more comprehensive analysis
To further comply with the reviewer’s requests, we also performed N-cadherin Western blot on Oz cells co-cultured with human Schwann cells. A trend increase (however not significant) of the mesenchymal marker N-cadherin was found as compared with single culture. We report such results in the Supplementary Figure 2. The following sentence has been added in paragraph 3.4:“An upregulation trend (yet statistically unsignificant) of the mesenchymal marker N-cadherin was found in Oz/hSCs indirect coculture as compared to single cultured Oz cells (Supplementary Fig. S2).”(lines 347-349, page 10).
Consistently with results obtained in HuCC-T1, also in Oz cells we found that HSCs CM increased BrdU incorporation respect to control. These results are now displayed in the new Figure 8. The following sentence has been added to paragraph 3.5:“In addition, proliferation was assessed in Oz at 48h of hSCs CM exposure considering that these cells have a longer doubling time respect to HuCC-T1.Remarkably,as already observed in HuCC-T1, also in Oz cells we found increased BrdU incorporation following treatment with hSCs CM respect to control CM(Fig. 8).”(lines 400-404, page 12)
Since clonogenic assay requires single-cell suspension for its plating, we could not perform it on Oz cells due to their high tendency to grow in clusters which are extremely difficult to disaggregate. In order to disaggregate Oz cells, trypsinization is not sufficient and several passages through 18 Ga needle are required, which result in extensive cell death (as observed via Trypan blue exclusion test performed during cell counts).
Question n. 4: Since IHC indicated lower CK7 levels in tumor cells invading nerves, the authors are encouraged to verify this observation by conducting Western blot analysis in HuCCT1 cells cocultured with human Schwann cells (hSCs).
Answer: We thank the reviewer for this comment. We performed Western blot analysis in HuCC-T1 cells cultured with hSCs CM. The results were included in supplementary Fig.S3, showing the down regulation of CK7 protein expression in cells exposed to hSCs. The following sentence has been added to paragraph 3.4:“ Since low CK7 immunohistochemistry positivity was detected in tumor cells invading nerves (Figure 1B), we performed Western blot analysis of CK7. The results reported in Fig. S3 show a decrease of CK7 protein in HuCC-T1 cells exposed to hSCs CM respect to control CM." (lines 350-353, page 10). Furthermore, the following sentence has been added to the Discussion: “This finding was confirmed in our in vitro model by Western blot analysis.” (line 477, page 15)
Question n. 5a: The list of both downregulated and upregulated proteins in HuCC-T1 single cultures versus HuCC-T1/hSCscocultures should be provided.
Answer: as requested, the complete list of both downregulated and upregulated proteins with the respective accession numbers and p values in HuCC-T1 single cultures versus HuCC-T1/hSCs cocultures has been provided in the supplementary Table 2. The following sentence has been introduced in paragraph 3.6: “The complete list of both downregulated and upregulated proteins with the respective accession numbers and p values in HuCC-T1 single cultures versus HuCC-T1/hSCs cocultures has been provided in the supplementary Table 2 (Tab. S.2)” (lines 423-425 page 13).
Question 5b: Additionally, the absence of p-values in Table 2 should be addressed and rectified.
Answer: As stated in the legend of Table 2, all the reported data have p<0.05.
Question n. 6: In Figure 9, it is recommended to perform the invasion assay for the Oz cell line to enhance the comprehensiveness of the experimental data.
Answer: As explained above (see answer to question 3a) non-significant results in invasion assays of Oz cells were collected; consequently, we did not perform invasion assays with the specific TGFBR1 antagonist.
Reviewer #3 minor concerns:
The quality of immunofluorescence images in Figures 3 and 4 should be improved, particularly for better visualization of nuclei under different culture conditions.
Answer: We thank for this comment; Figures 3 and 4 have been improved as requested.
The image quality of the Western blot in Figure 5 should be improved for better clarity and accuracy in data representation.
Answer: We thank for this comment; Figures 5 has been improved as requested.
Round 2
Reviewer 1 Report
Comments and Suggestions for Authors
All my concerns were addressed and the quality of this work was improved. Congratulations on this work.
Reviewer 2 Report
Comments and Suggestions for Authors
The authors responded satisfactorily to the questions posed by the reviewer. The manuscript is now significantly improved
Reviewer 3 Report
Comments and Suggestions for Authors
I thank the authors for following the various suggestions.